# Hibiscus Flower and Olive Leaf Extracts Activate Apoptosis in SH-SY5Y Cells

**DOI:** 10.3390/antiox10121962

**Published:** 2021-12-07

**Authors:** Elda Chiaino, Matteo Micucci, Roberta Budriesi, Laura Beatrice Mattioli, Carla Marzetti, Maddalena Corsini, Maria Frosini

**Affiliations:** 1Dipartimento di Scienze della Vita, Università di Siena, Via Aldo Moro 2, 53100 Siena, Italy; chiaino@student.unisi.it; 2Dipartimento di Farmacia e Biotecnologie, Alma Mater Studiorum-Università di Bologna, Via Belmeloro, 40126 Bologna, Italy; matteo.micucci2@unibo.it (M.M.); roberta.budriesi@unibo.it (R.B.); laura.mattioli13@unibo.it (L.B.M.); 3UniCamillus-Saint Camillus International University of Health Sciences, Via di Sant’Alessandro, 800131 Rome, Italy; 4Valsambro S.r.l., Via Cairoli 2, 40121 Bologna, Italy; carla.marzetti@valsambro.it; 5Dipartimento di Biotecnologie Chimica e Farmacia, Università di Siena, Via Aldo Moro 2, 53100 Siena, Italy; maddalena.corsini@unisi.it

**Keywords:** *Hibiscus sabdariffa* L., *Olea europea* L., natural compounds, neuroblastoma, apoptosis, cancer, caspases, oleuropein, hibiscus acid

## Abstract

Compounds of natural origin may constitute an interesting tool for the treatment of neuroblastoma, the most prevalent extracranial solid tumor in children. PRES is a commercially available food supplement, composed of a 13:2 (*w/w*) extracts mix of *Olea europaea* L. leaves (OE) and *Hibiscus sabdariffa* L. flowers (HS). Its potential towards neuroblastoma is still unexplored and was thus investigated in human neuroblastoma SH-SY5Y cells. PRES decreased the viability of cells in a concentration-dependent fashion (24 h IC_50_ 247.2 ± 31.8 µg/mL). Cytotoxicity was accompanied by an increase in early and late apoptotic cells (AV-PI assay) and sub G0/G1 cells (cell cycle analysis), ROS formation, reduction in mitochondrial membrane potential, and caspases activities. The ROS scavenger N-acetyl-L-cysteine reverted the cytotoxic effects of PRES, suggesting a key role played by ROS in PRES-mediated SH-SY5Y cell death. Finally, the effects of OE and HS extracts were singularly tested and compared to those of the corresponding mixture. OE- or HS-mediated cytotoxicity was always significantly lower than that caused by PRES, suggesting a synergic effect. In conclusion, the present findings highlight the potential of PRES for the treatment of neuroblastoma and offers the basis for a further characterization of the mechanisms underlying its effects.

## 1. Introduction

Advanced therapeutic strategies for cancer are mainly based on synthetic drugs and/or monoclonal humanized antibodies, that often produce severe adverse effects, despite designed for being selective. Consequently, the finding of non-toxic compounds, that still target particular genes or proteins should be a strategy that worth to be pursued, especially for pediatric cancer treatments. Neuroblastoma (NB), one of the most common extra-cranial solid cancer affecting children, is characterized by high clinical, biological, genetic, and morphological heterogeneity [1] and this makes it difficult to target for successful therapy [2]. It is responsible for up to 15% of the mortality associated to pediatric cancer and its overall survival rate is noticeably low (40% after 5 years) [3]. The diagnosis occurs mainly within the first year of age, thus resulting into a better outcome, whereas in the other patients the outcome itself is generally poor. NB treatment, despite some successes, remains highly demanding and is based on surgical resection, chemo- or radiation therapy. Unfortunately, surgery is not possible in more than 70% of patients as the disease is unresectable and metastatic, and thus multi-agent and multi-modal therapy remain the only options [2]. Chemotherapy is mainly based on a combination of drugs such as doxorubicin, vincristine, cyclophosphamide, etopomide, topetecan and cisplatin [4]. Although highly beneficial, it lacks selectivity and has severe side effects in both short and long terms, as survivors commonly have permanent health problems from the therapy received during childhood [5]. Unfortunately, NB incidence has increased over the last decade [6,7], and thus the therapy is under thorough investigation as there is an urgent need for drugs with greater efficacy to increase the survival rate, to improve the survivor’s quality of life and possibly to avoid a second malignancy.

Interestingly, compounds of natural origin have been proven to offer an attractive tool. Among their several biological activities, the possibility to target multiple cancer-mediated pathways at the same time and to modulate the complex tumor milieu constitute interesting properties to be explored. For example, several natural compounds such as resveratrol, curcumin and epigallocatechin-3-gallate (EGCG) were shown as promising in this respect [8]. Moreover, the combination of two or three plants with specific anticancer properties may addictively or even synergistically help classical chemotherapy by decreasing the doses needed, widening the efficacy window, overcoming multidrug resistance, and possibly lowering side effects resulting from high drug concentrations. The synergic effect of the combination of natural compounds has been investigated in multiple contexts both in vivo and in vitro [9,10] demonstrating how the exploration of natural therapeutics is continuously growing.

*Pres Phytum*^®^ (PRES) is a pharmaceutical formulation marketed as a food supplement composed by a 13:2 (*w/w*) mixture of the extracts obtained by *Olea europea* L. (OE) leaves and *Hybiscus sabdariffa* L. (HS) calyces, respectively. PRES possesses many biological properties, owing to the phenolic compounds found in both plants which includes oleuropein, hydroxytyrosol, flavonoids, chalcones, tannins, and hibiscus acid, as recently assessed [11,12]. As its potential towards NB is still unexplored, the aim of the present work was to investigate the cytotoxic effects of PRES in human neuroblastoma SH-SY5Y cells, focusing on its ability to induce cellular death and apoptosis, important hallmarks for potential anti-cancer therapies.

## 2. Materials and Methods

### 2.1. PRES

Detailed information about the preparation of OE- and HS-extracts, as well as the preparation of the mixture, were already reported [11,12]. Briefly, *Olea europea* L. leaves extract (EFLA^®^ 943, OE, Frutarom Switzerland Ltd., Wädenswil, Switzerland) was obtained by an ethanol (80% m/m) extraction. To remove contaminants and residues, the extract was purified by a patented procedure (U.S. Patent No. 6024998), which was followed by the removal of the solvent, thus resulting in a free-flowing powder. The *Hibiscus sabdariffa* L. flower powder extract (HS, Nutraceutica S.r.l., Monterenzio, Bologna, Italy) was obtained by dried calyces subjected to 48 h distilled water extraction, followed by filtration, concentration under reduced pressure, and complete evaporation by means of a vacuum oven kept at a temperature lower than 40 °C, and finally dried using a freeze-dryer system. HPLC coupled to a UV-Vis and QqQ-Ms detector was used for evaluating the main components of PRES. These were found to be (mg/g): in OE, 8.96 ± 0.4 hydroxyoleuropein, 23.65 ± 0.8 elenolic acid glucoside, 215.1 ± 1.6 oleuropein, 51.09 ± 0.3 oleuropein isomer, 11.03 ± 0.4 ligstroside, 4.67 ± 0.4 verbascoside, 0.69 ± 0.1 rutin, 0.74 ± 0.04 luteolin-7-*O*-rutinoside, 5.83 ± 0.2 luteolin-7-*O*-glucoside, 4.20 ± 0.1 luteolini-4-*O*-glucoside; and in HS, 139.2 ± 0.5 mg/g hibiscus acid [11,12].

### 2.2. Cell Cultures and PRES Treatments

Human SH-SY5Y neuroblastoma cells (passages 6–20, Cat# 94,030,304 ECACC), were cultured as previously reported [13,14], and experiments performed by using cells in the exponential phase of growth.

Stock solution of PRES (10 mg/mL in PBS, pH adjusted to 7.3, filtered by 0.45 µm pore size) was prepared immediately before use and diluted to the desired final concentration with cell culture medium. The stability of PRES solutions kept at different temperatures for 0–48 h were assessed by measuring UV-Vis spectra in the wavelength range of 200–600 nm in quartz cuvette with 1 cm optical path length (Multiskan TM GO, Thermo Fisher Scientific, Vantaa, Finland). No significant changes were detected in the spectra recorded, thus suggesting a good stability.

### 2.3. Cell Viability Assays and Assessment of Apoptosis

Cell viability was assessed by using MTT assay [15,16]. At the end of some experiments, a blind expert operator checked for PRES-induced SH-SY5Y changes in cell morphology by a phase-contrast microscope, and the observed cells injury was quantified in a score from 0 to 4 according to the USP 28 (United States Pharmacopeia edition 2005) grade scale [17,18]. To evaluate the role of ROS in the effects caused by PRES, N-acetyl-L-cysteine (NAC, 30 µM) was used. SH-SY5Y cells were thus pre-treated for 2 h with the ROS scavenger before adding the extract (250 or 500 µg/mL, 24 h) [17,18].

To check for cell apoptosis, SH-SY5Y flow cytometry sub-G0/G1 population and cell cycle analyses were performed [19] along with annexin V/PI double staining assay [17,19].

A FACScan flow cytometer (BD Biosciences, San Jose, CA, USA) and Cell Quest software v. 3.0 (BD Biosciences) were used to acquire and analyze samples, respectively [17,19].

### 2.4. Intracellular ROS Content, Mitochondrial Membrane Potential and Caspase Assays

Intracellular ROS formation and mitochondria integrity were investigated by using 2′, 7′-dichlorofluorescein diacetate (DCFH-DA), and rhodamine-123 (Rh123), respectively [17,20]. Specific caspase-3, 8 and 9 substrates, which release the fluorescence dye 7-amino-4-methylcoumarin (AMC, 380 nm excitation and 460 nm emission, SYNERGY HTX multi-mode reader, BioTek, Winooski, VT, USA) were used to assess for caspase activation [17].

### 2.5. Analysis of Data

Results were reported as means ± SEMs of at least three independent experiments run in qudruplicate. Cell viability resulting from MTT assay was reported as a percentage value of untreated cells (controls). Statistical significance was assessed by using ANOVA followed by the Dunnett or Bonferroni post hoc test, as appropriate (GraphPad Prism version 5.04, GraphPad Software Inc., San Diego, CA, USA). In all comparisons, the level of statistical significance (*p*) was set at 0.05.

## 3. Results

### 3.1. Effects of PRES on SH-SY5Y Cell Viability

Cytotoxic effects towards human neuroblastoma SH-SY5Y cells were assessed after 24 h of PRES treatment (0, 50, 100, 250, 500, 1000 µg/mL) by using an MTT assay. As reported in Figure 1a, the viability of treated cells was reduced as compared to that of cells not exposed to PRES. The effect was concentration-dependent and the IC_50_ value was 247.2 ± 31.8 µg/mL (*n* = 6, Appendix A). The smallest cytotoxic effect occurred at a 100 μg/mL and become more marked at higher concentrations, with the highest inhibition (~90%) observed at 1000 μg/mL. As previously mentioned, PRES is a 13:2 (*w/w*) mix of the extracts prepared from OE and HS, respectively. Thus, we found it of interest to assess whether the observed overall effect of PRES is a result of an additive (i.e., the combined effect is a pure summation effect), antagonistic (i.e., the combined effect is lower than the summation) or synergic (i.e., combined effect is greater than the additive) activity between OE and HS components. The effects of the corresponding amount found in 250 µg (217.0 µg of OE and 33.0 µg of HS) and 500 µg (434.0 µg of OE and 66.0 µg of HS) were singularly tested along with the resultant mixture (OE + HS, corresponding to PRES) on SH-SY5Y cell viability (Figure 1b). Interestingly, (OE + HS)-mediated cytotoxicity was always significantly higher than that caused by the single components (OE or HS). In particular, the latter were poorly active (OE) or ineffective (HS) when assessed singularly at 267.0 µg and 33.0 µg, respectively, while (OE + HS) 250 µg halved the number of viable cells, thus suggesting a synergistic effect. At the higher concentration, 434 µg OE reduced cells’ viability by ~52.0%, while 66 µg HS was still ineffective; however, the viability detected after 500 µg of (OE + HS) was significantly lower.

To highlight critical cellular modifications that might pass unnoticed using conventional cytotoxicity assays, analysis at contrast-phase microscopy was performed. The differences in cell survival detected by using MTT assay were paralleled by changes in cell morphology (Figure 2). Phase-contrast microscope analysis showed that the SH-SY5Y untreated cells grew normally as tight colonies (cytotoxicity grade 0) and at 100 µg/mL PRES, only few of them had irregular shape and appeared shrunken (cytotoxicity grade 1). On the contrary, cells treated with 250 and 500 µg/mL PRES presented important changes in both the number and shape. Wider spaces among cells in fact occurred, which was accompanied by shrinking, tendency detach from the well and to round-up (cytotoxicity grade 2–3), a morphology resembling the typical appearance of apoptotic cells.

### 3.2. PRES Induced an Apoptotic-Mediated SH-SY5Y Cell Death

#### 3.2.1. Cell Cycle Analysis

To further study the mechanisms causing cell death, flow cytometry-mediated cells cycle analysis was performed by using 100 µg/mL (the highest PRES safe concentration) along with 250 and 500 µg/mL (grade 2 or 3 of toxicity). As reported in Figure 3, an increase in the number of cells in sub G0/G1 phase was evident as PRES concentration increased, with significant values at 250 and 500 µg/mL, suggesting an occurring significant cell apoptosis. The increase in the sub-G0/G1 phase was accompanied by a decrease in the number of cells in the G0/G1 and G2/M phase, which is also an indication of cell apoptosis, while S cells were not affected.

#### 3.2.2. Annexin V/Propidium Iodide Staining

The occurrence of phosphatidyl serine externalization was also used to assess for apoptosis by using annexin V/PI assay. PRES caused a marked increase in early apoptotic (+14.3% and +22.9%, for 250 and 500 µg/mL PRES, respectively, *p* < 0.001 vs. control) and late apoptotic cells (+4.3% and +21.9%, for 250 µg/mL and 500 µg/mL PRES, respectively, *p* < 0.001 vs. control) which was accompanied by a gradual reduction in healthy cells (Figure 4). Finally, necrotic cells averaged~7% and were not changed by the treatments.

### 3.3. PRES Caused ROS Formation, Which Drives Cytotoxicity

ROS formation was investigated as many chemotherapy drugs cause cell apoptosis as a consequence of increased formation of ROS themselves, that in turn further stimulate cell apoptosis and DNA damage. Interestingly, the present findings showed that both 250 and 500 µg/mL PRES elicited a significant increase in intracellular DCF and thus in ROS formation (Figure 5a). To further support their role in the cytotoxic effects of PRES, the ROS scavenger N-acetyl-L-cysteine (NAC, 30 µM) was used. In this case, SH-SY5Y cells’ viability was significantly recovered (Figure 5b), thus suggesting that ROS drives PRES-mediated SH-SY5Y cell death.

### 3.4. PRES Increased the Cells with Loss in Mitochondria Membrane Potential

Changes in mitochondrial membrane integrity is considered to be one of the early events in apoptosis [21]. Thus, the mitochondria membrane potential (MMP) changes resulting from PRES treatment was investigated by means of Rh123. As reported in Figure 5c, cells with loss in MMP were comparable to untreated cells for 100 µg/mL PRES, while these increased significantly for both 250 and 500 µg/mL PRES concentrations.

### 3.5. PRES Affected also Caspase Activity

Mitochondria- and the extrinsic receptor-mediated pathways of apoptosis are both involved in the induction of cell death upon the activation of caspase-9 and 8, respectively. These, in turn, activate caspase-3 and fragmentation of DNA [21]. To further investigate which pathway(s) is (are) involved in the observed effects, fluorescence assays with specific caspase-3, 8 and 9 substrates were performed. As reported in Figure 6, the activities of all cleaved caspase-3, -8 and -9 were significantly increased, especially at the highest PRES concentration, suggesting the activation of both the mitochondrial- and extrinsic receptor-mediated apoptosis pathways.

## 4. Discussion

Many natural compounds are currently under investigation for treating pediatric cancer with the final aim to reduce side effects and possible the toxicity of the existing chemotherapeutic treatments [22]. Plant-derived drugs exert anticancer activity by many different mechanisms, including the promotion of apoptosis. In cancer, in fact, the apoptotic pathways are hampered by the under- or over-expression of proapoptotic or antiapoptotic proteins, respectively, thus resulting in the intrinsic resistance to the common chemotherapy [23]. Owing to their improved bioactivities and lower toxicity, a promising approach is to combine plant-derived compounds endowed with pro-apoptotic activity to the conventional therapy to enhance cancer cells’ susceptibility [22,24,25].

As detailed before, PRES is a commercially available nutraceutical product composed of leaf and flower extracts of OE and HS, the composition of which is characterized in detail [11,12]. Quantitatively the most important compounds are oleuropein-like compounds (hydroxyoleuropein, oleuropein itself and its isomer), and elenolic acid (hydroxyoleuropein, oleuropein itself and its isomer and elenolic acid glucoside; from OE leaves) and hibiscus acid (or (+)-hydroxycitric acid; from HS calyces). Recently, OE leaves, which constitute a non-edible, discarded part of the plant but still retaining high nutritional values, were proved to exert beneficial effects on human health, especially antitumoral [26,27]. Many reports, in fact, demonstrated that OE leaf extract might protect from several cancers [28] and that the polyphenols contained act synergistically [29], thus making the use of the extract more appropriate than that of the isolated compounds. Furthermore, OE leaves active components can preferentially target cancer- rather than normal-cells, rendering more interesting their use as co-adjuvant compounds [30,31]. On the other hand, also the anthocyanins found in HS calyces possess antitumoral properties [32] and although these constitute a minor part in PRES, might contribute to the effects. The present study represents the very first phase for exploring the potential of PRES for the treatment of neuroblastoma. Results seemed to be promising as the extracts caused a concentration-dependent decrease in cell viability (24 hour-IC_50_ ~250 µg/mL). A synergist effect was also demonstrated, as the cytotoxicity caused by the mixture was always significantly higher than that elicited by the single OE leaf and HS calyces’ extracts. This agrees with previous data which showed PRES exerting higher activities than each separate extract at cardiovascular level [11]. Additionally, morphologic and flow cytometric analysis revealed that PRES-induced cytotoxicity involved apoptosis-mediated cell death. Apoptotic-like death features observed by contrast-phase microscopy along with the occurrence of early and late apoptotic cells (AV-PI assay) and an increase in sub G0/G1 cells (cell cycle analysis) were in fact observed. Moreover, the present results suggest that the interesting activity of PRES was due to all its components and not only to oleuropein-like compounds, the most representative from a quantitative point of view. Data from the literature assessing OE leaf extract anti-tumor activity against different neuroblastoma cell lines, including SH-SY5Y, reported significant loss in cell viability and apoptosis after 72 h treatment with the extract containing 300 µM oleuropein [33]. Seçme et al. directly investigated the anti-tumor efficacy of the single bioactive molecule oleuropein against the SH-SY5Y neuroblastoma cell line, finding an approximate 48 hour-IC_50_ of 350 µM [34]. In the present study, the 24 hour-IC_50_ value of PRES was~250 µg/mL (corresponding to~100 µM oleuropein), while that of only OE leaves extract was~430 µg/mL (corresponding to 212 µM oleuropein). Taken together the present results highlighted a most potent activity of PRES respect to that of the single OE extract or of oleuropein, suggesting that also the components from HS calyces might play a role in the pro-apoptotic effects of PRES. In the literature, no data about the activity of HS extract on neuroblastoma are reported and thus it will be interesting to assess how its components (hibiscus acid?) promote synergic effects with those of OE leaf extract.

Mitochondria, being the main source of cellular ROS and ATP, play a crucial regulatory role in the mechanisms controlling the balance of cells survival/death pathways [21,23]. Moreover, beside phosphatidylserine externalization, changes in mitochondrial membrane potential constitute one of the most important hallmarks of the earliest intracellular events of apoptosis. Results showed that a significant rise in ROS formation occurred upon PRES treatment, indicating a pro-oxidant-mediated cytotoxic effect. This was accompanied by an increase in SH-SY5Y cells with loss of mitochondria membrane potential, suggesting that PRES-mediated apoptosis may be related to the activation of the mitochondrial-mediated pathway. The increase in ROS formation agrees with several reports demonstrating that plant-derived phytochemicals can enhance ROS production [35]. The pro-oxidant effect is becoming an interesting property strictly connected to the antitumor mechanisms especially because it seems to be one of the crucial steps triggering the apoptotic cascade. This is further supported by the observation of the present study, according to which the ROS scavenger NAC reverted the cytotoxic effects caused by PRES. Moreover, the most representative compound of PRES, oleuropein, can be a pro-oxidant both in vitro [36] and in cancer cell models [18], including SH-SY5Y cells [37], in the same range of concentrations used in the present study. As in cancer cells intracellular copper levels are elevated [38,39], we can speculate that a transfer of electrons between copper ions and PRES polyphenols occurs, giving rise to ROS. Cancer cells are under continuous oxidative stress as a consequence of increased growth rate and metabolism, thus resulting in depleted antioxidant systems [40]. Therefore, ROS generated by PRES polyphenols can overwhelm the SH-SY5Y cells’ antioxidant capacity, thus causing irreversible damage and apoptosis. This hypothesis is supported by the observation that the formation of copper-oleuropein complex plays a key role in the oleuropein-induced SH-SY5Y apoptosis, and it is reverted or increased when cells are treated with a copper chelator or non-toxic copper concentrations, respectively [41]. Finally, PRES-induced formation of ROS was comparable between 250 and 500 µg/mL, suggesting that a plateau was probably reached. This can be explained by considering that the copper-dependent ROS formation might be slowed down by a limited copper availability at the highest PRES concentration.

The effects of PRES on the key caspases involved in apoptosis, the initiator caspase 8 and caspase 9, and the executioner caspase 3, were also assessed. Results confirmed that PRES caused the activation of both the mitochondria dependent and extrinsic pathways, which involves the activation of the caspase-9 and caspase-8, respectively, and ultimately to caspase-3. Interestingly, the crosstalk between the extrinsic and intrinsic apoptosis pathways can result in an efficient induction in cell death, as demonstrated in several cancer cells [42,43,44], and upon the treatment with several natural compounds [45,46,47]. These results indicate that PRES can be considered promising in the research of new drugs for neuroblastoma treatment because it triggers at least two apoptosis pathways that could help to evade possible strategies of resistance activated by cancer cells. Moreover, many natural compounds target cancer cells sparing healthy tissue as oleuropein, whose toxic effect appears to be specific for the neuroblastoma rather than neuronal healthy cells [37]. Thus, we can speculate that also PRES might possess this selectivity, as suggested by data showing that PRES does not affect rat brain slices viability at least up to 200 µg/mL [18].

The possibility that PRES polyphenols may exert their biological effects towards neuroblastoma cells, however, depends on their bioavailability as well as their capability to access the brain. Regarding the ADME properties, data on plasma levels and/or excretion in urine of PRES main component oleuropein are conflicting, probably because many are the factors affecting its bioavailability [48]. Nevertheless, de Bock et al. showed that after oral ingestion of an OE leaves extract in men, oleuropein resulted to be resistant to the acidity of the stomach, was rapidly absorbed in the intestinal tract, reaching a peak plasma concentration in the order of ng/mL [49]. García-Villalba et al, found oleuropein urine metabolites at micromolar concentration in women supplemented with oleuropein-rich OE leaf extract [50]. Moreover, hibiscus acid, hibiscus acid hydroxyethyl ester, and the metabolite hydroxycitric acid reached µM plasma concentrations after oral ingestion of HS extract in rats [51]. In the light of these considerations, one critical point of the present in vitro study could be that the concentration of polyphenols used are in the µM range, at variance with those measured in plasma, which are generally lower. This highlights the need to direct more efforts to the characterization of the in vivo activity of PRES. However, it is worth noting that bioavailability might be enhanced by some minor components of the phytocomplex able to increase the absorption rate of the major constituents in the extract [52], or by a positive role exerted by the gut microbiota [46]. Future research will assess if this is the case of PRES. Moreover, several delivery systems have been recently developed to increase phenolic compounds’ bioavailability and interesting progress has been made in adopting pharmaceuticals oral delivery strategies including solubilizing technologies for small molecules, lipid-based systems, nanoparticles, intestinal permeation enhancers [53]. Finally, regarding the possibility to access the brain, an in-silico prediction of drug-like properties showed that oleuropein, elenolic acid glucoside and hibiscus acid, have good chances to be orally bioavailable and to cross the blood–brain barrier [18], as confirmed by the fact that oleuropein or hydroxytyrosol (its active metabolite), are both found in the brain in a significant amount following peripheral oleuropein administration [54,55].

## 5. Conclusions

Finding effective drugs for treating neuroblastoma is challenging. A possible approach is to use a multi-drug therapy able to act simultaneously on more than one target. In this respect, the use of natural compounds along with traditional chemotherapy might prove useful owing also to a possible synergism which might result in a better therapeutic efficacy. The present findings highlight interesting cytotoxic properties of PRES toward neuroblastoma, due also to a synergic effect among its main components. These might penetrate the brain, a factor which constitutes an added value to its potential use. As a result, additional research is needed to robustly demonstrate the clinical usefulness of PRES. The present findings reveal interesting cytotoxic properties of PRES towards neuroblastoma cells, probably due to a synergistic effect between its main components. However, a decisive test of whether or not they penetrate the brain would help determine their value for potential use.

## Figures and Tables

**Figure 1 antioxidants-10-01962-f001:**
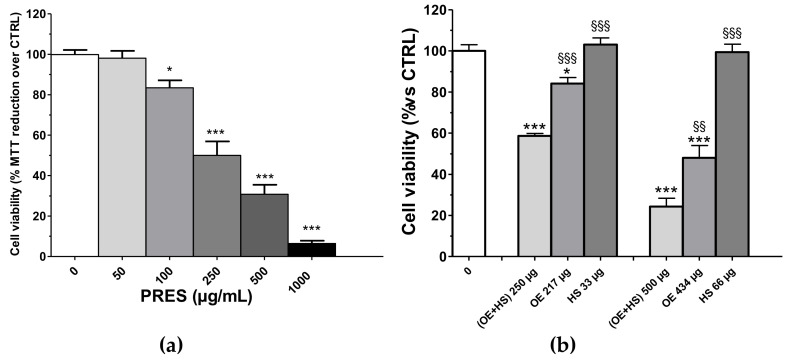
(**a**) Effects of PRES (0–1000 µg/mL, 24 h) on SH-SY5Y cell viability. Controls (CTRL; PRES 0 µg/mL) represent untreated cells. (**b**) SH-SY5Y cells’ viability after treatment with 250 µg/mL or 500 µg/mL of a 13:2 (*w/w*) mix of the extracts prepared from *Olea europea* L. leaves (OE) and *Hybiscus sabdariffa* L. calyces (HS), respectively, or to the corresponding amount found in 250 µg/mL or 500 µg/mL of only the *Olea europea* L. leaves extract (OE, 217 and 433 µg/mL, respectively) or *Hybiscus sabdariffa* L. calyces (HS, 33 and 67 µg/mL, respectively). Data were reported as mean ± SEMs. * *p* < 0.05, *** *p* < 0.001 vs. CTRL; §§ *p* < 0.01, §§§ *p* < 0.001 vs. (OE + HS) same bar group (ANOVA followed by Bonferroni post hoc test).

**Figure 2 antioxidants-10-01962-f002:**
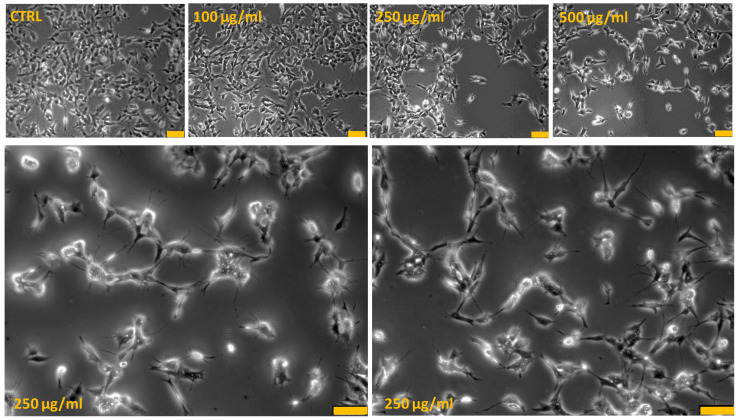
PRES-induced human neuroblastoma SH-SY5Y cells morphological changes: contrast-phase microscopy analysis. Bottom photos represent an enlarged view of 250 µg/mL PRES treated cells in which it is evident cells shrinkage and membrane bleb formations typically occurring in apoptotic-mediated cells death. Each photograph was representative of three independent observations (yellow line: scale bar 75 µm).

**Figure 3 antioxidants-10-01962-f003:**
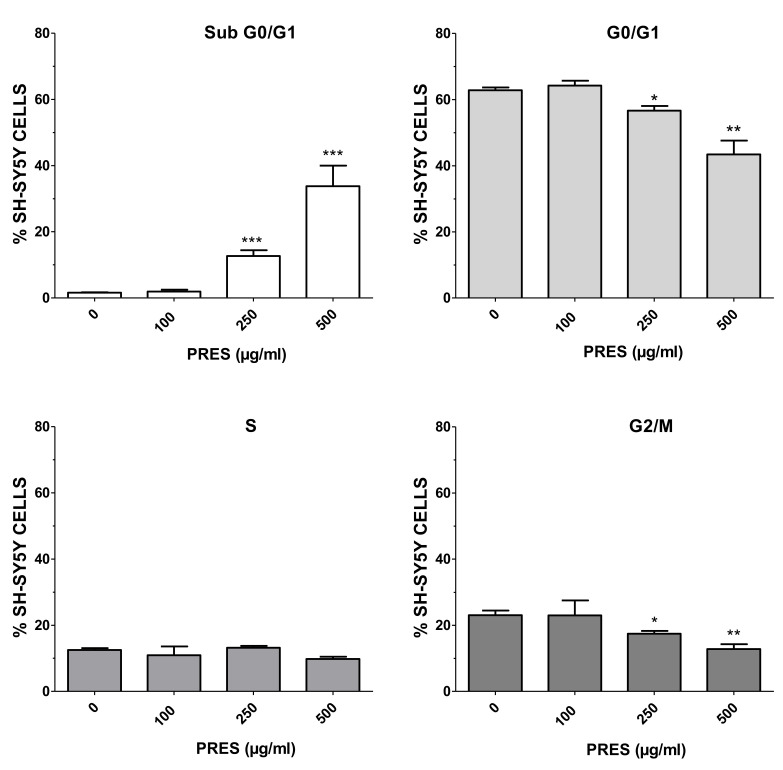
Percentage of SH-SY5Y cells at different phases of the cell cycle following treatment for 24 h with 100, 250 or 500 µg/mL PRES. Controls (CTRL, PRES 0 µg/mL) represent untreated cells. Data are reported as mean ± SEMs. * *p* < 0.05, ** *p* < 0.01, *** *p* < 0.001 vs. CTRL (ANOVA followed by Dunnet’s post hoc test).

**Figure 4 antioxidants-10-01962-f004:**
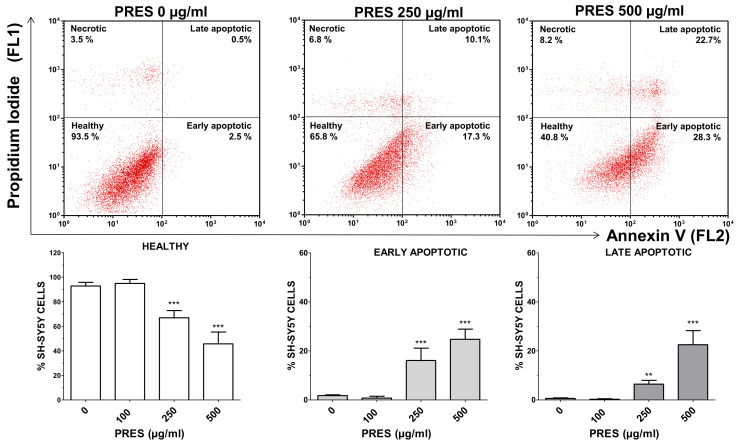
Apoptotic cells’ detection by double staining with annexin V (AV) and propidium iodide (PI). Top: representative dot plot of SH-SY5Y cells treated with 0, 250 or 500 µg/mL pf PRES, showing progressive increase in early- and late-apoptotic cells. Bottom: quantitative analysis. After the treatment with PRES (0, 100, 250 and 500 µg/mL for 24 h), SH-SY5Y cells were stained with AV and PI and flow cytometric analysis performed. Healthy cells were those negative for both AV and PI. Cells in early apoptosis were those stained by annexin V but not by PI, while those in late apoptosis were stained by both dyes. The excitation of the fluorochromes was performed at 488 nm, while the emitted green (AV) and red (PI) fluorescences were, respectively, detected at the wavelength of 530 ± 30 nm (FL1) and 585 ± 42 nm (FL2). Values are means ± SEMs; controls (CTRL, PRES 0 µg/mL) represent untreated cells. ** *p* < 0.01, *** *p* < 0.001 vs. CTRL (ANOVA followed by Dunnet’s post hoc test).

**Figure 5 antioxidants-10-01962-f005:**
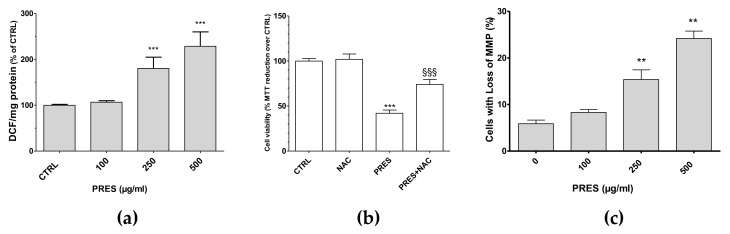
(**a**) PRES-mediated effects on intracellular ROS formation; (**b**) cell viability in the presence of N-acetyl-L-cysteine (NAC); (**c**) and cells with loss in MMP. Data are reported as means ± SEMs; controls (CTRL, PRES 0 µg/mL) represent untreated cells. ** *p* < 0.01, *** *p* < 0.001 vs. CTRL; §§§ *p* < 0.01 vs. PRES (ANOVA followed by Bonferroni post hoc test).

**Figure 6 antioxidants-10-01962-f006:**
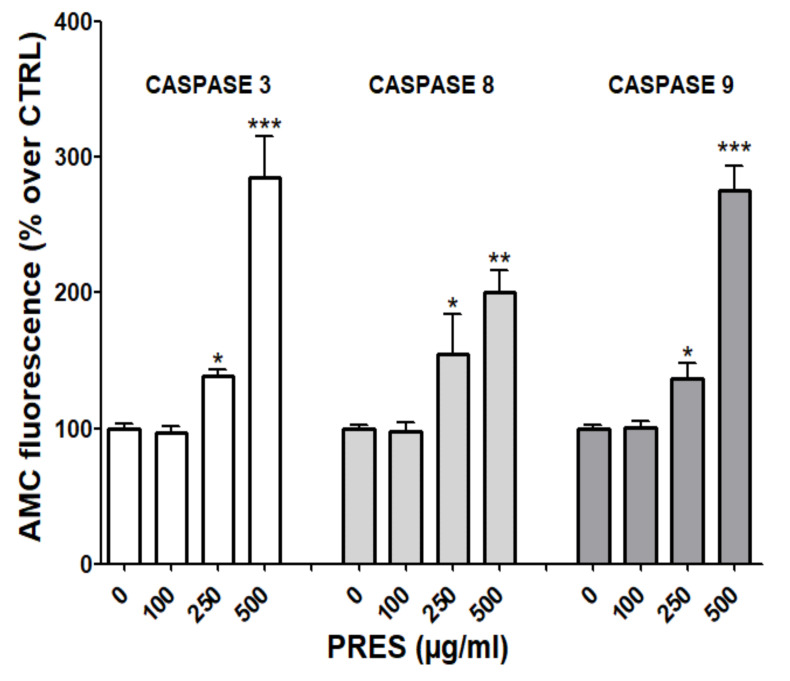
PRES induced caspase-3, 8 and 9 changes in human neuroblastoma SHSY-5Y cells. Data (means ± SEMs) are reported as percent values of the fluorescence of the AMC-fragment released by active caspases in untreated cells (PRES 0 µg/mL). * *p* < 0.05, ** *p* < 0.01, *** *p* < 0.001 vs. controls (ANOVA followed by Dunnet’s post hoc test).

## Data Availability

Data is contained within the article and Appendix A.

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
