# Peer review of "Hibiscus Flower and Olive Leaf Extracts Activate Apoptosis in SH-SY5Y Cells"

_antioxidants, 2021, doi:10.3390/antiox10121962_

Round 1
Reviewer 1 Report
In these well written manuscripts authors are presenting effect of Olea europaea L. leaves (OE) and Hibiscus sabdariffa extracts in treatment of neuroblastoma. Authors provide large number of experiments to support theirs statement. All experiments are well designed and described. Figures are very well presented and proper statistical analysis is applied. The evidence provided in paper is compelling and very interesting to the field.
The only one minor issue which I would address is Figure2, while changes of cells confluency are obvious, morphological changes describe by the authors as irregular shapes of cells and shrink like phenotype are hard to localize in current pictures. I think it would be beneficial to show higher magnification of those cells to show apoptotic phenotype of SH-SY5Y cells upon treatment with PRES.
Author Response
In these well written manuscripts authors are presenting effect of Olea europaea L. leaves (OE) and Hibiscus sabdariffa extracts in treatment of neuroblastoma. Authors provide large number of experiments to support theirs statement. All experiments are well designed and described. Figures are very well presented and proper statistical analysis is applied. The evidence provided in paper is compelling and very interesting to the field. The only one minor issue which I would address is Figure2, while changes of cells confluency are obvious, morphological changes describe by the authors as irregular shapes of cells and shrink like phenotype are hard to localize in current pictures. I think it would be beneficial to show higher magnification of those cells to show apoptotic phenotype of SH-SY5Y cells upon treatment with PRES.
We thank the Referee for his suggestion. In the revised version of the manuscript, Figure 2 has been rearranged providing enlarged view photos of SH-SY5Y cells treated with 250 µg/ml PRES.
Reviewer 2 Report
Materials and methods line 81 - the standard of most research and analytical works is to describe the experience in such a way that the potential reader can repeat it. therefore, it is necessary to add OE and HS ex-entries. I can see that the authors indicate the source of their research [11, 12] but you need to know that not every reader has access to various publications.
the same note for graphs (not diagrams) Figure 4. Apoptotic
It would be reasonable to present the independent action on OE and HS cancer cells, and as in the work of their synergism, it would show even more how they interact more strongly. I know, of course, that the volume of the work is limited, but I am very curious about it, so I ask if such research has been done.
The authors are aware of the need for research on the possible implementation of the therapy with the use of the discussed extracts, or is the kilnic approach the next stage of the planned research?
Figure 3. Percentage..for better readability of the presented results, all four graphs should have the same value of the OY axis .. if the authors state that the last two (s and GM / 2) should remain to the value of 40, then those above should at least unify
Author Response
Ref2
Materials and methods line 81 - the standard of most research and analytical works is to describe the experience in such a way that the potential reader can repeat it. therefore, it is necessary to add OE and HS ex-entries. I can see that the authors indicate the source of their research [11, 12] but you need to know that not every reader has access to various publications.
We thank the Referee for his suggestion. We omitted the details about OE and HS preparation as these are available in a free full-text quoted reference. We agree however that these details could be of interest for the reader and thus have now included more details as follows:
Detailed information about the preparation of OE- and HS-extracts, as well as the preparation of the mixture, were already reported [11,12]. Briefly, Olea europea L. leaves extract (EFLA® 943, OE, Frutarom Switzerland Ltd., Wädenswil, Switzerland) was obtained by an ethanol (80% m/m) extraction. To remove contaminants and residues, the extract was purified by a patented procedure (US Patent 6024998), which was followed by the removal of the solvent, thus resulting in a free-flowing powder. The Hibiscus sabdariffa L. flower powder extract (HS, Nutraceutica S.r.l., Monterenzio, Bologna, Italy) was obtained by dried calyces subjected to 48-h distilled water extraction, followed by filtration, concentration under reduced pressure, and complete evaporation by means of a vacuum oven kept at a temperature lower than 40°C, and finally dried using Freeze Dryer system. For more de-tails, visit http://www.nutraceutica.it
The same note for graphs (not diagrams) Figure 4. Apoptotic
We agree with Referee that the legend of bottom panels of Figure 4 was excessively synthetic. In the revised manuscript more details were included as follows:
Bottom: quantitative analysis. After the treatment with PRES (0, 100, 250 and 500 µg/ml for 24h), SH-SY5Y cells were stained with AV and PI and flow cytometric analysis performed. Healthy cells were those negative for both AV and PI. Cells in early apoptosis were those stained by annexin V but not by PI, while those in late apoptosis were stained by both dyes. The excitation of the fluorochromes was performed at 488 nm, while the green (AV) and red (PI) fluorescence emitted were respectively detected at the wavelength of 530 ± 30 nm (FL1) and 585 ± 42 nm (FL2).
It would be reasonable to present the independent action on OE and HS cancer cells, and as in the work of their synergism, it would show even more how they interact more strongly. I know, of course, that the volume of the work is limited, but I am very curious about it, so I ask if such research has been done.
The research was performed and was indeed already reported in the manuscript (see Figure 1, panel b). The effects of the corresponding amount found in 250 µg (217.0 µg of OE and 33.0 µg of HS) and 500 µg (434.0 µg of OE and 66.0 µg of HS) of PRES were singularly tested along with the resultant mixture (OE + HS, corresponding to PRES itself) on SH-SY5Y cell viability. Results showed that (OE + HS)-mediated cytotoxicity was always significantly higher than that caused by the single components (OE or HS), thus suggesting a synergistic effect.
The authors are aware of the need for research on the possible implementation of the therapy with the use of the discussed extracts, or is the kilnic approach the next stage of the planned research?
We are entirely aware that the present study represents the very first phase for exploring the potential of PRES for the treatment of neuroblastoma. As outlined also in the discussion, one critical point of the present in vitro study could be that the concentration of polyphenols used are in the µM range, at variance with those measured in plasma, which are generally lower. This highlights the need to direct more efforts to the characterization of the in vivo activity of PRES (next stage of our research), although data from literature suggest that oleuropein, elenolic acid glucoside and hibiscus acid, have good chances to be orally bioavailable and to cross the blood-brain barrier (see refs 18, 54, and 55). Finally, as reported also in the manuscript, the use of natural compounds such as PRES, should be seen as a support of traditional chemotherapy, with the aim to possibly enhance cancer cells’ susceptibility.
Figure 3. Percentage..for better readability of the presented results, all four graphs should have the same value of the OY axis .. if the authors state that the last two (s and GM / 2) should remain to the value of 40, then those above should at least unify
We agree with the Referee that it worth to unify 0Y axis scale of panels in Figure 3. This has been performed by choosing 0-80 as the scale for all the four panels.
Reviewer 3 Report
The perspectives on the potential antitumoral effects of PRES are, probably, overstated. Taking into account the scientific profile of the Journal, it would be clearly better to reorient the article towards a deeper explanation of the oxidative processes within the mitochondria. The fact that oleuropein is considered as a pro-oxidant seems contradictory with some previous results (e. g., Antioxidants 10(2):328 (2021) Polyphenols in the mediterranean diet: From dietary sources to microRNA modulation. Roberto Cannataro , Alessia Fazio , Chiara La Torre, Maria Cristina Caroleo , Erika Cione). For that, an explanation is needed.
Some minor details are included in the attached pdf file.

Author Response
The perspectives on the potential antitumoral effects of PRES are, probably, overstated. Taking into account the scientific profile of the Journal, it would be clearly better to reorient the article towards a deeper explanation of the oxidative processes within the mitochondria. The fact that oleuropein is considered as a pro-oxidant seems contradictory with some previous results (e. g., Antioxidants 10(2):328 (2021) Polyphenols in the mediterranean diet: From dietary sources to microRNA modulation. Roberto Cannataro , Alessia Fazio , Chiara La Torre, Maria Cristina Caroleo , Erika Cione). For that, an explanation is needed.
We agree with the Referee that oleuropein pro-oxidant activity seems to be in contrast to its well-known antioxidant properties. Phenolic compounds, however, can act as antioxidants or pro-oxidants, depending on their concentration and the cellular environment. This pro-oxidant effect s is becoming an innovative finding closely linked to the antitumor mechanisms exerted by these compounds, which has additionally been hypothesized to be one of the key points triggering the apoptotic process in cancer cells (ref 35). The pro-oxidants activity is typically catalysed by metals, particularly transition metals such as Fe and Cu, present in biological systems, especially in tumoral cells (see for ex. PMID 27241122, 26206193, 21776260 …). This hold true also for oleuropein, that behave as a pro-oxidant both in vitro [ref 36] and in cancer cell models [ref 18], including SH-SY5Y cells [ref 37], the same cell line of the present study. About the central role of the oxidative processes at mitochondrial level, this was already outlined in lines 327-355 of the manuscript. The possible mechanisms underlying the pro-oxidant activity of PRES were in fact discussed, reporting also comparative data from the literature about oleuropein. The pro-oxidant activity seems to be a key point to explain its pro-apoptotic activity and is probably correlated to high copper ions content present in cancer cells (ref 38, 39) which possesses depleted antioxidant systems (ref 40).
Some minor details are included in the attached pdf file.
Minor details have been all amended.
Round 2
Reviewer 3 Report
Conclusions
1. Written idea: The present findings highlight interesting cytotoxic properties of PRES toward neuroblastoma, due also to a synergic effect among its main components. These might penetrate the brain, a factor which constitutes an added value to its potential use.
2. New proposed idea: The present findings reveal interesting cytotoxic properties of PRES towards neuroblastoma cells, probably due to a synergistic effect between its main components. However, a decisive test of whether or not they penetrate the brain would help determine their value for potential use.
References: I think all common nouns and adjectives that appear in the title of a given article should be written completely in lower case. Therefore, the authors should make the necessary changes.
All the scientific botanical names (e.g. Olea europaea) must be written in italics.
Author Response
- 1. Written idea: The present findings highlight interesting cytotoxic properties of PRES toward neuroblastoma, due also to a synergic effect among its main components. These might penetrate the brain, a factor which constitutes an added value to its potential use.
- New proposed idea: The present findings reveal interesting cytotoxic properties of PRES towards neuroblastoma cells, probably due to a synergistic effect between its main components. However, a decisive test of whether or not they penetrate the brain would help determine their value for potential use.
We fully agree with the Referee to re-write conclusions, as we overestimate the potential of PRES. The sentence has been changed as suggested.
References: I think all common nouns and adjectives that appear in the title of a given article should be written completely in lower case. Therefore, the authors should make the necessary changes.
We apologize for this negligence in quoting References. As we used the the editing program Mendeley, we thought that these were correct. All the inconsistences have been properly amended.
All the scientific botanical names (e.g. Olea europaea) must be written in italics.
Changes have been performed accordingly.
This manuscript is a resubmission of an earlier submission. The following is a list of the peer review reports and author responses from that submission.